# BYZANTINE-ROBUST LEARNING ON HETEROGENEOUS DATASETS VIA BUCKETING

**Sai Praneeth Karimireddy**[*]
sai.karimireddy@epfl.ch

**Lie He**[*]
lie.he@epfl.ch

**Martin Jaggi**
martin.jaggi@epfl.ch

## ABSTRACT

In Byzantine robust distributed or federated learning, a central server wants to train a machine learning model over data distributed across multiple workers. However, a fraction of these workers may deviate from the prescribed algorithm and send arbitrary messages. While this problem has received significant attention recently, most current defenses assume that the workers have identical data. For realistic cases when the data across workers are heterogeneous (non-iid), we design new attacks which circumvent current defenses, leading to significant loss of performance. We then propose a simple bucketing scheme that adapts existing robust algorithms to heterogeneous datasets at a negligible computational cost. We also theoretically and experimentally validate our approach, showing that combining bucketing with existing robust algorithms is effective against challenging attacks. Our work is the first to establish guaranteed convergence for the non-iid Byzantine robust problem under realistic assumptions.

## 1 INTRODUCTION

Distributed or federated machine learning, where the data is distributed across multiple workers, has become an increasingly important learning paradigm both due to growing sizes of datasets, as well as data privacy concerns. In such a setting, the workers collaborate to train a single model without directly transmitting their training data (McMahan et al., 2016; Bonawitz et al., 2019; Kairouz et al., 2019). However, by decentralizing the training across a vast number of workers we potentially open ourselves to new security threats. Due to the presence of agents in the network which are actively malicious, or simply due to system and network failures, some workers may disobey the protocols and send arbitrary messages; such workers are also known as *Byzantine* workers (Lamport et al., 2019). Byzantine robust optimization algorithms attempt to combine the updates received from the workers using robust aggregation rules and ensure that the training is not impacted by the presence of a small number of malicious workers.

While this problem has received significant recent attention due to its importance, (Blanchard et al., 2017; Yin et al., 2018; Alistarh et al., 2018; Karimireddy et al., 2021), most of the current approaches assume that the data present on each different worker has identical distribution. This assumption is very unrealistic in practice and heterogeneity is inherent in distributed and federated learning (Kairouz et al., 2019). In this work, we show that existing Byzantine aggregation rules catastrophically fail with very simple attacks (or sometimes even with no attacks) in realistic settings. We carefully examine the causes of these failures, and propose a simple solution which provably solves the Byzantine resilient optimization problem under heterogeneous workers.

Concretely, our contributions in this work are summarized below

- We show that when the data across workers is heterogeneous, existing aggregation rules fail to converge, even when no Byzantine adversaries are present. We also propose a simple new attack, *mimic*, which explicitly takes advantage of data heterogeneity and circumvents median-based defenses. Together, these highlight the fragility of existing methods in real world settings.

- We then propose a simple fix - a new bucketing step which can be used before any existing aggregation rule. We introduce a formal notion of a robust aggregator (ARAGG) and prove that existing methods like KRUM, coordinate-wise median (CM), and geometric median aka

---

[*]equal contribution.

robust federated averaging (RFA)—though insufficient on their own—become provably robust aggregators when augmented with our bucketing.

- We combine our notion of robust aggregator (ARAGG) with worker momentum to obtain optimal rates for Byzantine robust optimization with matching lower bounds. Unfortunately, our lower bounds imply that convergence to an exact optimum may not be possible due to heterogeneity. We then circumvent this lower bound and show that when heterogeneity is mild (or when the model is overparameterized), we can in fact converge to an exact optimum. This is the first result establishing convergence to the optimum for heterogeneous Byzantine robust optimization.

- Finally, we evaluate the effect of the proposed techniques (bucketing and worker momentum) against known and new attacks showcasing drastic improvement on realistic heterogeneously distributed datasets.

**Setup and notations.** Suppose that of the total $n$ workers, the set of good workers is denoted by $\mathcal{G} \subseteq \{1, \ldots, n\}$. Our objective is to minimize

$$f(\boldsymbol{x}) := \frac{1}{|\mathcal{G}|} \sum_{i \in \mathcal{G}} \left\{ f_i(\boldsymbol{x}) := \mathbb{E}_{\boldsymbol{\xi}_i}[F_i(\boldsymbol{x}; \boldsymbol{\xi}_i)] \right\} \tag{1}$$

where $f_i$ is the loss function on worker $i$ defined over its own (heterogeneous) data distribution $\boldsymbol{\xi}_i$. The (stochastic) gradient computed by a good worker $i \in \mathcal{G}$ over minibatch $\boldsymbol{\xi}_i$ is given as $\boldsymbol{g}_i(\boldsymbol{x}, \boldsymbol{\xi}_i) := \nabla F_i(\boldsymbol{x}; \boldsymbol{\xi}_i)$. The noise in every stochastic gradient is independent, unbiased with $\mathbb{E}_{\boldsymbol{\xi}_i}[\boldsymbol{g}_i(\boldsymbol{x}, \boldsymbol{\xi}_i)] = \nabla f_i(\boldsymbol{x})$, and has bounded variance $\mathbb{E}_{\boldsymbol{\xi}_i} \|\boldsymbol{g}_i(\boldsymbol{x}, \boldsymbol{\xi}_i) - \nabla f_i(\boldsymbol{x})\|^2 \leq \sigma^2$. Further, we assume that the data heterogeneity across the workers can be bounded as

$$\mathbb{E}_{j \sim \mathcal{G}} \|\nabla f_j(\boldsymbol{x}) - \nabla f(\boldsymbol{x})\|^2 \leq \zeta^2, \quad \forall \boldsymbol{x} \,.$$

We write $\boldsymbol{g}_i^t$ or simply $\boldsymbol{g}_i$ instead of $\boldsymbol{g}_i(\boldsymbol{x}^t, \boldsymbol{\xi}_i^t)$ when there is no ambiguity.

**Byzantine attack model.** The set of Byzantine workers $\mathcal{B} \subset [n]$ is fixed over time, with the remaining workers $\mathcal{G}$ being good, i.e. $[n] = \mathcal{B} \uplus \mathcal{G}$. We write $\delta$ for the fraction of Byzantine workers, $|\mathcal{B}| =: q \leq \delta n$. The Byzantine workers can deviate arbitrarily from our protocol, sending any update to the server. Further, they can collude and may even know the states of all other workers.

Our modeling assumes that the practitioner picks a value of $\delta \in [0, 0.5)$. This $\delta$ reflects the level of robustness required. A choice of a large $\delta$ (say near 0.5) would mean that the system is very robust and can tolerate a large fraction of attackers, but the algorithm becomes much more conservative and slow. On the flip side, if the practitioner knows that the the number of Byzantine agents are going to be few, they can pick a small $\delta$ (say 0.05–0.1) ensuring some robustness with almost no impact on convergence. The choice of $\delta$ can also be formulated as how expensive do we want to make an attack? To carry out a succesful attack the attacker would need to control $\delta$ fraction of all workers. We recommend implementations claiming robustness be transparent about their choice of $\delta$.

## 2   RELATED WORK

**IID defenses.** There has been a significant amount of recent work on the case when all workers have identical data distributions. Blanchard et al. (2017) initiated the study of Byzantine robust learning and proposed a distance-based aggregation approach KRUM and extended to (El Mhamdi et al., 2018; Damaskinos et al., 2019). Yin et al. (2018) propose to use and analyze the coordinate-wise median (CM), and Pillutla et al. (2019) use approximate geometric median. Bernstein et al. (2018) propose to use the signs of gradients and then aggregate them by majority vote, however, Karimireddy et al. (2019) show that it may fail to converge. Most recently, Alistarh et al. (2018); Allen-Zhu et al. (2021); El Mhamdi et al. (2021b); Karimireddy et al. (2021) showcase how to use past gradients to more accurately filter iid Byzantine workers and specifically *time-coupled* attacks. In particular, our work builds on top of (Karimireddy et al., 2021) and non-trivially extends to the non-iid setting.

**IID vs. Non-IID attacks.** For the iid setting, the state-of-the-art attacks are *time-coupled* attacks (Baruch et al., 2019; Xie et al., 2020). These attacks introduce a small but consistent bias at every step which is hard to detect in any particular round, but accumulates over time and eventually leads to divergence, breaking most prior robust methods. Our work focuses on developing attacks (and defenses) which specifically take advantages of the non-iid setting. The non-iid setting also enables targeted *backdoor* attacks which are designed to take advantage of heavy-tailed data (Bagdasaryan et al., 2020; Bhagoji et al., 2018). However, this is a challenging and open problem (Sun et al., 2019; Wang et al., 2020). Our focus is on the overall accuracy of the trained model, not on any subproblem.

**Non-IID defenses.** The non-iid defenses are relatively under-examined. Ghosh et al. (2019); Sattler et al. (2020) use an outlier-robust clustering method. When the server has the entire training dataset, the non-iid-ness is automatically addressed (Xie et al., 2019; Chen et al., 2018; Rajput et al., 2019). Typical examples are parallel training of neural networks on public cloud, or volunteer computing (Meeds et al., 2015; Miura & Harada, 2015). Note that Rajput et al. (2019) use hierarchical aggregation over "vote group" which is similar to the bucketing techniques but their results are limited to the iid setting. However, none of these methods are applicable to the standard federated learning. This is partially tackled in (Data & Diggavi, 2020; 2021) who analyze spectral methods for robust optimization. However, these methods require $\Omega(d^2)$ time, making them infeasible for large scale optimization. Li et al. (2019) proposes an SGD variant (RSA) with additional $\ell_p$ penalty which only works for strongly convex objectives. In an independent recent work, Acharya et al. (2021) analyze geometric median (GM) on non-iid data using sparsified gradients. However, they do not defend against time coupled attacks, and their analysis neither proves convergence to the optimum nor recovers the standard rate of SGD when $\delta \to 0$. In contrast, our analysis of GM addresses both issues and is more general. For decentralized training with non-iid data, a parallel work (El Mhamdi et al., 2021a) considers asynchronous communication and unconstrained topologies and tolerates a maximum number of Byzantine workers in their setting. However, no convergence rate is given. He et al. (2022) consider decentralized training on constrained topologies and establish the consensus and convergence theory for a clipping based algorithm which tolerates a $\delta$-fraction of Byzantine workers, limited by the spectral gap of the topology. Finally, Yang & Li (2021) propose to use bucketing for asynchronous Byzantine learning which is very similar to the bucketing trick proposed in this paper for non-iid setup. In Appendix F.1 we further compare these two methods in terms of purposes, techniques, and analysis. [1]

**Strong growth condition.** The assumption that $\mathbb{E}_{j\sim\mathcal{G}}\|\nabla f_j(\boldsymbol{x}) - \nabla f(\boldsymbol{x})\|^2 \leq B^2\|\nabla f(\boldsymbol{x})\|^2$ for some $B \geq 0$ is also referred to as the strong growth condition (Schmidt & Roux, 2013). This has been extensively used to analyze and derive optimization algorithms for deep learning (Schmidt & Roux, 2013; Ma et al., 2018; Vaswani et al., 2019a;b; Meng et al., 2020). This line of work shows that the strong growth assumption is both realistic and (perhaps more importantly) useful in understanding optimization algorithms in deep learning. However, this is stronger than the *weak* growth condition which states that $\mathbb{E}_{j\sim\mathcal{G}}\|\nabla f_j(\boldsymbol{x}) - \nabla f(\boldsymbol{x})\|^2 \leq B^2(f(\boldsymbol{x}) - f^\star)$ for some $B \geq 0$. For a smooth function $f$, the strong growth condition always implies the weak growth condition. Further, for smooth convex functions this is equivalent to assuming that all the workers functions $\{f_i\}$ share a common optimum, commonly known as interpolation. Our work uses the stronger version of the growth condition and it remains open to extend our results to the weaker version. This latter condition is strictly necessary for heterogeneous Byzantine optimization (Gupta & Vaidya, 2020).

## 3 ATTACKS AGAINST EXISTING AGGREGATION SCHEMES

In this section we show that when the data across the workers is heterogeneous (non-iid), then we can design simple new attacks which take advantage of the heterogeneity, leading to the failure of existing aggregation schemes. We study three representative and widely used defenses:

**Krum.** For $i \neq j$, let $i \to j$ denote that $\boldsymbol{x}_j$ belongs to the $n - q - 2$ closest vectors to $\boldsymbol{x}_i$. Then,

$$\mathrm{KRUM}(\boldsymbol{x}_1, \ldots, \boldsymbol{x}_n) := \operatorname*{argmin}_i \sum_{i \to j} \|\boldsymbol{x}_i - \boldsymbol{x}_j\|^2 .$$

Krum is computationally expensive, requiring $\mathcal{O}(n^2)$ work by the server (Blanchard et al., 2017).
**CM.** Coordinate-wise median computes for the $k$th coordinate:

$$[\mathrm{CM}(\boldsymbol{x}_1, \ldots, \boldsymbol{x}_n)]_k := \mathrm{median}([\boldsymbol{x}_1]_k, \ldots, [\boldsymbol{x}_n]_k) = \operatorname*{argmin}_i \sum_{j=1}^n |[\boldsymbol{x}_i]_k - [\boldsymbol{x}_j]_k| .$$

Coordinate-wise median is fast to implement requiring only $\mathcal{O}(n)$ time (Chen et al., 2017).

**RFA.** Robust federated averaging (RFA) computes the geometric median

$$\mathrm{RFA}(\boldsymbol{x}_1, \ldots, \boldsymbol{x}_n) := \operatorname*{argmin}_{\boldsymbol{v}} \sum_{i=1}^n \|\boldsymbol{v} - \boldsymbol{x}_i\|_2 .$$

While the geometric median has no closed form solution, (Pillutla et al., 2019) approximate it using multiple iterations of smoothed Weiszfeld algorithm, each of which requires $\mathcal{O}(n)$ computation.

---

[1]The previous version of this work uses resampling which has identical performance as bucketing. The detailed comparison is listed in Appendix A.2.4.

Table 1: Test accuracy (%) with no Byzantine workers ($\delta = 0$) on imbalanced data.

| Aggr | iid | non-iid |
|------|-----|---------|
| Avg | 98.79±0.10 | 98.75±0.02 |
| Krum | 97.95±0.25 | 89.90±4.75 |
| CM | 97.72±0.22 | 80.36±0.05 |
| RFA | 98.62±0.08 | 82.60±0.84 |
| CClip | 98.78±0.10 | 98.78±0.06 |

Table 2: Test accuracy (%) under mimic attack with $\delta = 0.2$ fraction of Byzantine workers.

| Aggr | iid | non-iid |
|------|-----|---------|
| Avg | 93.20±0.21 | 92.73±0.32 |
| Krum | 90.36±0.25 | 37.33±6.78 |
| CM | 90.80±0.12 | 64.27±3.70 |
| RFA | 92.92±0.25 | 78.93±9.27 |
| CClip | 93.16±0.22 | 91.53±0.06 |

### 3.1 FAILURE ON IMBALANCED DATA WITHOUT BYZANTINE WORKERS

We show that when the data amongst the workers is imbalanced, existing aggregation rules *fail* even in the *absence* of any Byzantine workers. Algorithms like KRUM select workers who are *representative* of a majority of the workers by relying on statistics such as pairwise differences between the various worker updates. Our key insight is that when the data across the workers is heterogeneous, there is no single worker who is representative of the whole dataset. This is because each worker computes their local gradient over vastly different local data.

**Example.** Suppose that there are $2n + 1$ workers with worker $i$ holding $(-1)^i \in \{\pm 1\}$. This means that the true mean is $\approx 0$, but KRUM, CM, and RFA will output $\pm 1$. This motivates our next attack.

Hence, for convergence it is important to not only select a good (non-Byzantine) worker, but also ensure that each of the good workers is selected with roughly equal frequency. In Table 1, we demonstrate failures of such aggregators by training on MNIST with $n = 20$ and no attackers ($\delta = 0$). We construct an imbalanced dataset where each successive class has only a fraction of samples of the previous class. We defer details of the experiments to Appendix A. As we can see, KRUM, CM and RFA match the ideal performance of SGD in the iid case, but only attain less than 90% accuracy in the non-iid case. This corresponds to learning only the top 2–3 classes and ignoring the rest.

A similar phenomenon was observed when using batch-size 1 in the iid case by (Karimireddy et al., 2021). However, in the iid case this can be easily overcome by increasing the batch-size. In contrast, when the data across the works is non-iid (e.g. split by class), increasing the batch-size does *not* make the worker gradients any more similar and there remains a big drop in performance. Finally, note that in Table 1 a hitherto new algorithm (CCLIP) maintains its performance both in the iid and the non-iid setting. We will explore this in more detail in Section 4.

### 3.2 MIMIC ATTACK ON BALANCED DATA

Motivated by how data imbalance could lead to consistent errors in the aggregation rules and significant loss in accuracy, in this section, we will propose a new attack *mimic* which specifically tries to maximize the perceived data imbalance even if the original data is balanced.

**Mimic attack.** All Byzantine workers pick a good worker (say $i_\star$) to mimic and copy its output ($\boldsymbol{x}_{i_\star}^t$). This inserts a consistent bias towards over-emphasizing worker $i_\star$ and thus under-representing other workers. Since the attacker simply mimics a good worker, it is impossible to distinguish it from a real worker and hence it cannot be filtered out. Indeed, the target $i_\star$ can be any fixed good worker. In Appendix B, we present an empirical rule to choose $i_\star$ and include a simple example demonstrating how median based aggregators suffer from the heterogeneity under mimic attack.

Table 2 shows the effectiveness of mimic attack even when the fraction of Byzantine nodes is small (i.e. $n = 25$, $|\mathcal{B}| = 5$). Note that this attack specifically targets the non-iid nature of the data— all robust aggregators maintain their performance in the iid setting and only suffer in the non-iid setting. Their performance is in fact worse than even simply averaging. As predicted by our example, KRUM and CM have the worst performance and RFA performs slightly better. We will discuss the remarkable performance of CCLIP in the next section.

## 4 CONSTRUCTING AN AGNOSTIC ROBUST AGGREGATOR USING BUCKETING

In Section 3 we demonstrated how existing aggregation rules fail in realistic non-iid scenarios, with and without attackers. In this section, we show how using bucketing can provably fix such aggregation rules. The underlying reason for this failure, as we saw previously, is that the existing methods fixate on the contribution of only the most likely worker, and ignore the contributions from the rest. To

---

**Algorithm 1** Robust Aggregation (ARAGG) using bucketing

---
1: **input** $\{\boldsymbol{x}_1, \ldots, \boldsymbol{x}_n\}$, $s \in \mathbb{N}$, aggregation rule AGGR
2: pick random permutation $\pi$ of $[n]$
3: compute $\boldsymbol{y}_i \leftarrow \frac{1}{s} \sum_{k=(i-1)\cdot s+1}^{\min(n,\, i\cdot s)} \boldsymbol{x}_{\pi(k)}$ for $i = \{1, \ldots, \lceil n/s \rceil\}$
4: **output** $\hat{\boldsymbol{x}} \leftarrow \text{AGGR}(\boldsymbol{y}_1, \ldots, \boldsymbol{y}_{\lceil n/s \rceil})$         // aggregate after bucketing

---

overcome this issue, we propose to use bucketing which 'mixes' the data from all the workers thereby reducing the chance of any subset of the data being consistently ignored.

### 4.1 BUCKETING ALGORITHM

Given $n$ inputs $\boldsymbol{x}_1, \ldots, \boldsymbol{x}_n$, we perform *s-bucketing* which randomly partitions them into $\lceil n/s \rceil$ buckets with each bucket having no more than $s$ elements. Then, the contents of each bucket are averaged to construct $\{\boldsymbol{y}_1, \ldots, \boldsymbol{y}_{\lceil n/s \rceil}\}$ which are then input to an aggregator AGGR. The details are summarized in Algorithm 1. The key property of our approach is that after bucketing, the resulting set of averaged $\{\boldsymbol{y}_1, \ldots, \boldsymbol{y}_{\lceil n/s \rceil}\}$ are much more homogeneous (lower variance) than the original inputs. Thus, when fed into existing aggregation schemes, the chance of success increases. We formalize this in the following simple lemma.

**Lemma 1** (Bucketing reduces variance). *Suppose we are given $n$ independent (but not identical) random vectors $\{\boldsymbol{x}_1, \ldots, \boldsymbol{x}_n\}$ such that a good subset $\mathcal{G} \subseteq [n]$ of size at least $|\mathcal{G}| \geq n(1 - \delta)$ satisfies:*
$$\mathbb{E}\|\boldsymbol{x}_i - \boldsymbol{x}_j\|^2 \leq \rho^2, \quad \text{for any fixed } i, j \in \mathcal{G}.$$

*Define $\bar{\boldsymbol{x}} := \frac{1}{|\mathcal{G}|} \sum_{j \in \mathcal{G}} \boldsymbol{x}_j$. Let the outputs after s-bucketing be $\{\boldsymbol{y}_1, \ldots, \boldsymbol{y}_{\lceil n/s \rceil}\}$ and denote $\tilde{\mathcal{G}} \subseteq \{1, \ldots, \lceil n/s \rceil\}$ as a good bucket set where a good bucket contains only elements belonging to $\mathcal{G}$. Then $|\tilde{\mathcal{G}}| \geq \lceil n/s \rceil(1 - \delta s)$ satisfies*
$$\mathbb{E}[\boldsymbol{y}_i] = \mathbb{E}[\bar{\boldsymbol{x}}] \quad \text{and} \quad \mathbb{E}\|\boldsymbol{y}_i - \boldsymbol{y}_j\| \leq \rho^2/s \quad \text{for any fixed } i, j \in \tilde{\mathcal{G}}.$$

The expectation in the above lemma is taken both over the random vectors as well as over the randomness of the bucketing procedure.

**Remark 2.** *Lemma 1 proves that after our bucketing procedure, we are left with outputs $\boldsymbol{y}_i$ which have i) pairwise variance reduced by $s$, and ii) potentially $s$ times more fraction of Byzantine vectors. Hence, bucketing trades off increasing influence of Byzantine inputs against having more homogeneous vectors. Using $s = 1$ simply shuffles the inputs and leaves them otherwise unchanged.*

### 4.2 AGNOSTIC ROBUST AGGREGATION

We now define what it means for an agnostic robust aggregator to succeed.

**Definition A** (($\delta_{\max}, c$)-ARAGG). *Suppose we are given input $\{\boldsymbol{x}_1, \ldots, \boldsymbol{x}_n\}$ of which a subset $\mathcal{G}$ of size at least $|\mathcal{G}| > (1 - \delta)n$ for $\delta \leq \delta_{\max} < 0.5$ and satisfies $\mathbb{E}\|\boldsymbol{x}_i - \boldsymbol{x}_j\|^2 \leq \rho^2$. Then, the output $\hat{\boldsymbol{x}}$ of a Byzantine robust aggregator satisfies:*
$$\mathbb{E}\|\hat{\boldsymbol{x}} - \bar{\boldsymbol{x}}\|^2 \leq c\delta\rho^2 \quad \text{where} \quad \hat{\boldsymbol{x}} = \text{ARAGG}_\delta(\boldsymbol{x}_1, \ldots, \boldsymbol{x}_n).$$
*Further, ARAGG does not need to know $\rho^2$ (only $\delta$), and automatically adapts to any value $\rho^2$.*

Our robust aggregator is parameterized by $\delta_{\max} < 0.5$ which denotes the maximum amount of Byzantine inputs it can handle, and a constant $c$ which determines its performance. If $\delta = 0$, i.e. when there are no Byzantine inputs, we are guaranteed to *exactly* recover the true average $\bar{\boldsymbol{x}}$. Exact recovery is also guaranteed when $\rho = 0$ since in that case it is easy to identify the good inputs since they are all equal and in majority. When both $\rho > 0$ and $\delta > 0$, we recover the average up to an additive error term. We also require that the robust aggregator is *agnostic* to the value of $\rho^2$ and automatically adjusts its output to the current $\rho$ during training. The aggregator can take $\delta$ as an input though. This property is very useful in the context of Byzantine robust optimization since the variance $\rho^2$ keeps changing over the training period, whereas the fraction of Byzantine workers $\delta$ remains constant. This is a major difference from the definition used in (Karimireddy et al., 2021). Note that Definition A is defined for both homogeneous and heterogeneous data.

We next show that aggregators which we saw were not robust in Section 3, can be made to satisfy Definition A by combining with bucketing.

**Theorem I.** *Suppose we are given $n$ inputs $\{\boldsymbol{x}_1, \ldots, \boldsymbol{x}_n\}$ satisfying properties in Lemma 1 for some $\delta \leq \delta_{\max}$, with $\delta_{\max}$ to be defined. Then, running Algorithm 1 with $s = \lfloor \delta_{\max}/\delta \rfloor$ yields the following:*

- *Krum:* $\mathbb{E}\|\text{KRUM} \circ \text{BUCKETING}(\boldsymbol{x}_1, \ldots, \boldsymbol{x}_n) - \bar{\boldsymbol{x}}\|^2 \leq \mathcal{O}(\delta\rho^2)$ *with* $\delta_{\max} < 1/4$.
- *Geometric median:* $\mathbb{E}\|\text{RFA} \circ \text{BUCKETING}(\boldsymbol{x}_1, \ldots, \boldsymbol{x}_n) - \bar{\boldsymbol{x}}\|^2 \leq \mathcal{O}(\delta\rho^2)$ *with* $\delta_{\max} < 1/2$.
- *Coordinate-wise median:* $\mathbb{E}\|\text{CM} \circ \text{BUCKETING}(\boldsymbol{x}_1, \ldots, \boldsymbol{x}_n) - \bar{\boldsymbol{x}}\|^2 \leq \mathcal{O}(d\delta\rho^2)$ *with* $\delta_{\max} < 1/2$.

Note that all these methods satisfy our notion of an *agnostic* Byzantine robust aggregator (Definition A). This is because both our bucketing procedures as well as the underlying aggregators are independent of $\rho^2$. Further, our error is $\mathcal{O}(\delta\rho^2)$ and is information theoretically optimal, unlike previous analyses (e.g. Acharya et al. (2021)) who had an error of $\mathcal{O}(\rho^2)$.

The error of CM depends on the dimension $d$ which is problematic when $d \gg n$. However, we suspect this is because we measure stochasticity using Euclidean norms instead of coordinate-wise. In practice, we found that CM often outperforms KRUM, with RFA outperforming them both. Note that we select $s = \lfloor \delta_{\max}/\delta \rfloor$ to ensure that after bucketing, we have the maximum amount of Byzantine inputs tolerated by the method with $(s\delta) = \delta_{\max}$.

**Remark 3** (1-step Centered clipping). *The 1-step centered clipping aggregator (CCLIP) given a clipping radius $\tau$ and an initial guess $\boldsymbol{v}$ of the average $\bar{\boldsymbol{x}}$ performs*

$$\text{CCLIP}(\boldsymbol{x}_1, \ldots, \boldsymbol{x}_n) = \boldsymbol{v} + \frac{1}{n}\sum_{i\in[n]}(\boldsymbol{x}_n - \boldsymbol{v})\min(1, \tau/\|\boldsymbol{x}_n - \boldsymbol{v}\|_2).$$

*Karimireddy et al. (2021) prove that CCLIP even without bucketing satisfies Definition A with $\delta_{\max} = 0.1$, and $c = \mathcal{O}(1)$. This explains its good performance on non-iid data in Section 3. However, CCLIP is not agnostic since it requires clipping radius $\tau$ as an input which in turn depends on $\rho^2$. Devising a version of CCLIP which automatically adapts its clipping radius is an important open question. Empirically however, we observe that simple rules for setting $\tau$ work quite well—we always use $\tau = \frac{10}{1-\beta}$ in our limited experiments where $\beta$ is the coefficient of momentum.*

While we have shown how to construct a robust aggregator which satisfies some notion of a robustness, we haven't yet seen how this affects the Byzantine robust *optimization* problem. We investigate this question theoretically in the next section and empirically in Section 6.

## 5 ROBUST NON-IID OPTIMIZATION USING A ROBUST AGGREGATOR

In this section, we study the problem of optimization in the presence of Byzantine workers and heterogeneity, given access to any robust aggregator satisfying Definition A. We then show that data heterogeneity makes Byzantine robust optimization especially challenging and prove lower bounds for the same. Finally, we see how mild heterogeneity, or sufficient overparameterization can circumvent these lower bounds, obtaining convergence to the optimum.

---

**Algorithm 2** Robust Optimization using any Agnostic Robust Aggregator

---

**Input:** ARAGG, $\eta$, $\beta$
1: **for** $t = 1, \ldots$ **do**
2:      **for** worker $i \in [n]$ **in parallel**
3:          $\boldsymbol{g}_i \leftarrow \nabla F_i(\boldsymbol{x}, \boldsymbol{\xi}_i)$ and $\boldsymbol{m}_i \leftarrow (1-\beta)\boldsymbol{g}_i + \beta\boldsymbol{m}_i$            // worker momentum
4:          **send** $\boldsymbol{m}_i$ if $i \in \mathcal{G}$, else send $*$ if Byzantine
5:      $\hat{\boldsymbol{m}} = \text{ARAGG}(\boldsymbol{m}_1, \ldots, \boldsymbol{m}_n)$ and $\boldsymbol{x} \leftarrow \boldsymbol{x} - \eta\hat{\boldsymbol{m}}$.     // update params using robust aggregate

---

### 5.1 ALGORITHM DESCRIPTION

In Section 4 we saw that bucketing could tackle heterogeneity across the workers by reducing $\zeta^2$. However, there still remains variance $\sigma^2$ in the gradients within each worker since each worker uses stochastic gradients. To reduce the effect of this variance, we rely on worker momentum. Each worker sends their local worker momentum vector $\boldsymbol{m}_i$ to be aggregated by ARAGG instead of $\boldsymbol{g}_i$:

$$\boldsymbol{m}_i^t = \beta\boldsymbol{m}_i^{t-1} + (1-\beta)\boldsymbol{g}_i(\boldsymbol{x}^{t-1}) \quad \text{for every } i \in \mathcal{G},$$
$$\boldsymbol{x}^t = \boldsymbol{x}^{t-1} - \eta\text{ARAGG}(\boldsymbol{m}_1^t, \ldots, \boldsymbol{m}_n^t).$$

This is equivalent to the usual momentum description up to a rescaling of step-size $\eta$. Intuitively, using worker momentum $\boldsymbol{m}_i$ averages over $1/(1-\beta)$ independent stochastic gradients $\boldsymbol{g}_i$ and thus reduces the effect of the within-worker-variance $\sigma^2$ (Karimireddy et al., 2021). Note that the resulting $\{\boldsymbol{m}_i\}$ are *still heterogeneous* across the workers. This heterogeneity is the key challenge we face.

## 5.2 Convergence rates

We now turn towards proving convergence rates for our bucketing aggregation method Algorithm 1 based on any existing aggregator AGGR. We will assume that for any fixed $i \in \mathcal{G}$

$$\mathbb{E}_{\boldsymbol{\xi}_i} \|\boldsymbol{g}_i(\boldsymbol{x}) - \nabla f_i(\boldsymbol{x})\|^2 \leq \sigma^2 \text{ and } \mathbb{E}_{j \sim \mathcal{G}} \|\nabla f_j(\boldsymbol{x}) - \nabla f(\boldsymbol{x})\|^2 \leq \zeta^2, \quad \forall \boldsymbol{x}. \qquad (2)$$

This first condition bounds the variance of the stochastic gradient within a worker whereas the latter is a standard measure of inter-client heterogeneity in federated learning (Yu et al., 2019; Khaled et al., 2020; Karimireddy et al., 2020). Under these conditions, we can prove the following.

**Theorem II.** *Suppose we are given a $(\delta_{\max}, c)$-ARAGG satisfying Definition A, and $n$ workers of which a subset $\mathcal{G}$ of size at least $|\mathcal{G}| \geq n(1 - \delta)$ faithfully follow the algorithm for $\delta \leq \delta_{\max}$. Further, for any good worker $i \in \mathcal{G}$ let $f_i$ be a possibly non-convex function with $L$-Lipschitz gradients, and the stochastic gradients on each worker be independent, unbiased and satisfy (2). Then, for $F^0 := f(\boldsymbol{x}^0) - f^\star$, the output of Algorithm 2 satisfies*

$$\frac{1}{T} \sum_{t=1}^{T} \mathbb{E}\|\nabla f(\boldsymbol{x}^{t-1})\|^2 \leq \mathcal{O}\left(c\delta\zeta^2 + \sigma\sqrt{\frac{LF^0}{T}(c\delta + 1/n)} + \frac{LF^0}{T}\right).$$

**Remark 4** (Unified proofs). *Remark 3 shows that CCLIP is a robust aggregator, and Theorem I shows KRUM, RFA, and CM on combining with sufficient bucketing are all robust aggregators satisfying Definition A. Most of these methods had no end-to-end convergence guarantees prior to our results. Thus, Theorem II gives the first unified analysis in both the iid and non-iid settings.*

When $\delta \to 0$ i.e. as we reduce the number of Byzantine workers, the above rate recovers the optimal $\mathcal{O}(\frac{\sigma}{\sqrt{Tn}})$ rate for non-convex SGD and even has linear speed-up with respect to the $n$ workers. In contrast, all previous algorithms for non-iid data (e.g. (Data & Diggavi, 2021; Acharya et al., 2021)) do not improve their rates for decreasing values of $\delta$. This is also empirically reflected in Section 3.1, where these algorithms are shown to fail even in the absence of Byzantine workers ($\delta = 0$).

Further, when $\zeta = 0$ the rate above simplifies to $\mathcal{O}(\frac{\sigma}{\sqrt{T}} \cdot \sqrt{c\delta + 1/n})$ which matches the iid Byzantine robust rates of (Karimireddy et al., 2021). In both cases we converge to the optimum and can make the gradient arbitrarily small. However, when $\delta > 0$ and $\zeta > 0$, Theorem II only shows convergence to a radius of $\mathcal{O}(\sqrt{\delta}\zeta)$ and not to the actual optimum. We will next explore this limitation.

## 5.3 Lower bounds and the challenge of heterogeneity

Suppose worker $j$ sends us an update which looks 'weird' and is very different from the updates from the rest of the workers. This may be because worker $j$ might be malicious and their update represents an attempted attack. It may also be because worker $j$ has highly *non-representative data*. In the former case the update should be ignored, whereas in the latter the update represents a valuable source of specialized data. However, it is impossible for the server to distinguish between the two situations. The above argument can in fact be formalized to prove the following lower bound.

**Theorem III.** *Given any optimization algorithm ALG, we can find $n$ functions $\{f_1(x), \ldots, f_n(x)\}$ of which at least $(1 - \delta)n$ are good (belong to $\mathcal{G}$), 1-smooth, $\mu$-strongly convex functions, and satisfy $\mathbb{E}_{i \sim \mathcal{G}} \|\nabla f_i(x) - \nabla f(x)\|^2 \leq \zeta^2$ such that the output of ALG has an error at least*

$$\mathbb{E}[f(\text{ALG}(f_1, \ldots, f_n)) - f^\star] \geq \Omega\left(\frac{\delta\zeta^2}{\mu}\right) \quad \text{and} \quad \mathbb{E}\|\nabla f(\text{ALG}(f_1, \ldots, f_n))\|^2 \geq \Omega(\delta\zeta^2).$$

The expectation above is over the potential randomness of the algorithm. This theorem unfrotunately implies that it is impossible to converge to the true optimum in the presence of Byzantine workers. Note that the above lower bound is information theoretic in nature and is independent of how many gradients are computed or how long the algorithm is run.

**Remark 5** (Matches lower bound). *Suppose that we satisfy the heterogeneity condition (2) with $\zeta^2 > 0$ and $\sigma = 0$. Then, the rate in Theorem II can be simplified to $\mathcal{O}(\delta\zeta^2 + 1/T)$. While the second term in this decays to 0 with $T$, the first term remains, implying that we only converge to a radius of $\sqrt{\delta}\zeta$ around the optimum. However, this matches our lower bound result from Theorem III and hence is in general unimprovable.*

This is a very strong negative result and seems to indicate that Byzantine robustness might be impossible to achieve in real world federated learning. This would be major stumbling block for deployment since the system would provably be vulnerable to attackers. We will next carefully examine the lower bound and will attempt to circumvent it.

Table 3: Table 1 + Bucketing (s=2).

| Aggr | iid | non-iid |
|------|-----|---------|
| AVG | 98.80±0.10 | 98.74±0.02 |
| KRUM | 98.35±0.20 | 93.27±0.10 |
| CM | 98.26±0.22 | 95.59±0.89 |
| RFA | 98.75±0.14 | 97.34±0.58 |
| CCLIP | 98.79±0.10 | 98.75±0.02 |

Table 4: Table 2 + Bucketing (s=2).

| Aggr | iid | non-iid |
|------|-----|---------|
| AVG | 93.17±0.23 | 92.67±0.27 |
| KRUM | 91.64±0.30 | 53.15±3.96 |
| CM | 91.91±0.24 | 78.60±3.15 |
| RFA | 93.00±0.23 | 91.17±0.51 |
| CCLIP | 93.17±0.23 | 92.56±0.21 |

## 5.4 CIRCUMVENTING LOWER BOUNDS USING OVERPARAMETERIZATION

We previously saw some strong impossibility results posed by heterogeneity. In this section, we show that while indeed in the worst case being robust under heterogeneity is impossible, we may still converge to the true optimum under more realistic settings. We consider an alternative bound of (2):

$$\mathbb{E}_{j\sim\mathcal{G}}\|\nabla f_j(\boldsymbol{x}) - \nabla f(\boldsymbol{x})\|^2 \leq B^2\|\nabla f(\boldsymbol{x})\|^2, \quad \forall \boldsymbol{x}. \tag{3}$$

Note that at the optimum $\boldsymbol{x}^\star$ we have $\nabla f(\boldsymbol{x}^\star) = 0$, and hence this assumption implies that $\nabla f_j(\boldsymbol{x}^\star) = 0$ for all $j \in \mathcal{G}$. This is satisfied if the model is *sufficiently over-parameterized* and typically holds in most realistic settings (Vaswani et al., 2019a).

**Theorem IV.** *Suppose we are given a $(\delta_{\max}, c)$-ARAGG and $n$ workers with loss functions $\{f_1, \ldots, f_n\}$ satisfying the conditions in Theorem II with $\delta \leq \delta_{\max}$ and (3) for some $B^2 < \frac{1}{60c\delta}$. Then, for $F^0 := f(\boldsymbol{x}^0) - f^\star$, the output of Algorithm 2 satisfies*

$$\frac{1}{T}\sum_{t=1}^{T}\mathbb{E}\|\nabla f(\boldsymbol{x}^{t-1})\|^2 \leq \mathcal{O}\left(\frac{1}{1-60c\delta B^2}\cdot\left(\sigma\sqrt{\frac{LF^0}{T}(c\delta + 1/n)} + \frac{LF^0}{T}\right)\right).$$

**Remark 6** (Overparameterization fixes convergence). *The rate in Theorem IV not only goes to 0 with T, but also matches that of the optimal iid rate of $\mathcal{O}(\frac{\sigma}{\sqrt{T}} \cdot \sqrt{c\delta + 1/n})$ (Karimireddy et al., 2021). Thus, using a stronger heterogeneity assumption allows us to circumvent lower bounds for the non-iid case and converge to a good solution even in the presence of Byzantine workers. This is the first result of its kind, and takes a major step towards realistic and practical robust algorithms.*

In the overparameterized setting, we can be sure that we will able to *simultaneously* optimize all worker's losses. Hence, over time the agreement between all worker's gradients increases. This in turn makes any attempts by the attackers to derail training stand out easily, especially towards the end of the training. To take advantage of this increasing closeness, we need an aggregator which automatically adapts the quality of its output as the good workers get closer. Thus, the *agnostic* robust aggregator is crucial to our overparameterized convergence result. We empirically demonstrate the effects of overparameterization in Appendix A.2.3.

## 6 EXPERIMENTS

In this section, we demonstrate the effects of bucketing on datasets distributed in a non-iid fashion. Throughout the section, we illustrate the tasks, attacks, and defenses by an example of training an MLP on a heterogeneous version of the MNIST dataset (LeCun et al., 1998). The dataset is sorted by labels and sequentially divided into equal parts among good workers; Byzantine workers have access to the entire dataset. Implementations are based on PyTorch (Paszke et al., 2019) and will be made publicly available.[2] We defer details of setup, implementation, and runtime to Appendix A.

**Bucketing against the attacks on non-iid data.** In Section 3 we have presented how heterogeneous data can lead to failure of existing robust aggregation rules. Here we apply our proposed bucketing with $s\!=\!2$ to the same aggregation rules, showing that bucketing overcomes the described failures. Results are presented in Table 3. Comparing Table 3 with Table 1, bucketing improves the aggregators' top-1 test accuracy on long-tail and non-iid dataset by 4% to 14% and allows them to learn classes at the tail distribution. For non-iid balanced dataset, bucketing also greatly improves the performance of KRUM and CM and makes RFA and CCLIP close to ideal performance. Similarly, combining aggregators with bucketing also performs much better on non-iid dataset under mimic attack. In Table 4, RFA and CCLIP recover iid accuracy, and KRUM, and CM are improved by around 15%.

**Bucketing against general Byzantine attacks.** In Figure 1, we present thorough experiments on non-iid data over 25 workers with 5 Byzantine workers, under different attacks. In each subfigure, we compare an aggregation rule with its variant with bucketing. The aggregation rules compared are KRUM, CM, RFA, CCLIP. 5 different kinds of attacks are applied (one per column in the figure): bit flipping (BF), label flipping (LF), *mimic* attack, as well as inner product manipulation (IPM) attack (Xie et al., 2020) and the "a little is enough" (ALIE) attack (Baruch et al., 2019).

---

[2] The code is available at this url.

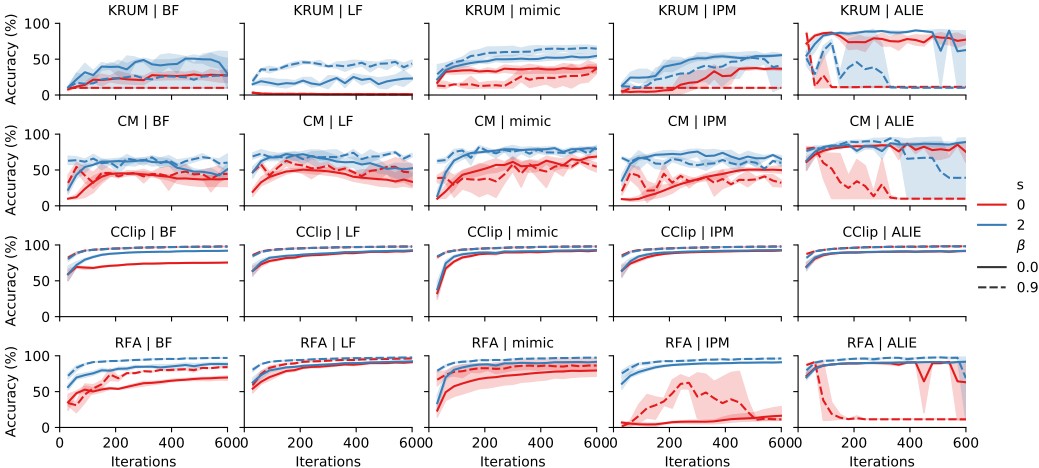

Figure 1: Top-1 test accuracies of KRUM, CM, CCLIP, RFA, under 5 attacks on non-iid datasets.

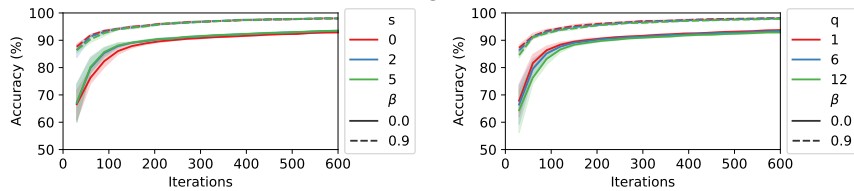

(a) Fixed $q = 5$ IPM attackers, varying $s$      (b) Fixed $s = 2$, varying number $q$

Figure 2: Top-1 accuracies of CCLIP with varying $q$ and $s$ when training on a cluster of $n = 53$ nodes

- **Bit flipping**: A Byzantine worker sends $-\nabla f(\boldsymbol{x})$ instead of $\nabla f(\boldsymbol{x})$ due to hardware failures etc.
- **Label flipping**: Corrupt MNIST dataset by transforming labels by $\mathcal{T}(y) := 9 - y$.
- **Mimic**: Explained in Section 3.2.
- **IPM**: The attackers send $-\frac{\epsilon}{|\mathcal{G}|} \sum_{i \in \mathcal{G}} \nabla f(\boldsymbol{x}_i)$ where $\epsilon$ controls the strength of the attack.
- **ALIE**: The attackers estimate the mean $\mu_{\mathcal{G}}$ and standard deviation $\sigma_{\mathcal{G}}$ of the good gradients, and send $\mu_{\mathcal{G}} - z\sigma_{\mathcal{G}}$ to the server where $z$ is a small constant controlling the strength of the attack.

Both IPM and ALIE are the state-of-the-art attacks in the iid distributed learning setups which takes advantage of the variances among workers. These attacks are much stronger in the non-iid setup. In the last two columns of Figure 1 we show that worker momentum and bucketing reduce such variance while momentum alone is not enough. Overall, Figure 1 shows that bucketing improves the performances of almost all aggregators under all kinds of attacks. Note that $\tau$ of CCLIP is not finetuned for each attack but rather fixed to $\frac{10}{1-\beta}$ for all attacks. This scaling is required because CCLIP is *not agnostic*. We defer the discussion to Appendix A.2.1.

**Bucketing hyperparameter.** Finally we study the influence of $s$ and $q$ on the heterogeneous MNIST dataset. We use CCLIP as the base aggregator and apply IPM attack. The Figure 2a confirms that larger $s$ gives faster convergence but $s = 2$ is sufficient. Figure 2b shows that $s = 2$ still behaves well when increasing $q$ close to 25%. The complete evaluation of the results are deferred to Appendix A.

**Discussion.** In all our experiments, we consistently observe: i) mild bucketing ($s = 2$) improves performance, ii) worker momentum further stabilizes training, and finally iii) CCLIP recovers the ideal performance. Given its ease of implementation, this leads us to strongly recommend using CCLIP in practical federated learning to safeguard against actively malicious agents or passive failures. RFA combined with bucketing and worker momentum also nearly recovers ideal performance and can instead be used when a proper radius $\tau$ is hard to find. Designing an *automatic* and *adaptively* clipping radius as well as its large scale empirical study is left for future work.

# 7 CONCLUSION

Heterogeneity poses unique challenges for Byzantine robust optimization. The first challenge is that existing defenses attempt to pick a "representative" update, which may not exist in the non-iid setting. This, we showed, can be overcome by using bucketing. A second more fundamental challenge is that it is difficult to distinguish between a "weird" but good worker from an actually Byzantine attacker. In fact, we proved strong impossibility results in such a setting. For this we showed how overparameterization (which is prevalent in real world deep learning) provides a solution, ensuring convergence to the optimum even in the presence of attackers. Together, our results yield a practical provably Byzantine robust algorithms for the non-iid setting.

**Acknowledgement.** This project was supported by SNSF grant 200020_200342.

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
