# OpenReview forum: "Byzantine-Robust Learning on Heterogeneous Datasets via Bucketing"
_ICLR.cc/2022/Conference — ICLR 2022 Spotlight_

### Official Review · Reviewer_ago1 · 2021-11-01

**Correctness:** 3
**Technical Novelty And Significance:** 3
**Empirical Novelty And Significance:** 3
**Recommendation:** 10
**Confidence:** 5

**Main Review:**

## Strengths
1. **Lower bounds for Byzantine-robust optimization under bounded heterogeneity.** In Theorem III, the authors prove that even in the case of bounded heterogeneity (in the classical sense for papers on FL) one cannot achieve any predefined accuracy even for the strongly convex problems. In particular, they prove that functional suboptimality cannot be made smaller than $\Omega(\frac{\delta \zeta^2}{\mu})$ ($\mu$ - str. convexity parameter, $\delta$ - fraction of Byzantines, $\zeta^2$ - dissimilarity measure of the gradients of local loss functions). Typically, $\mu$ is quite small and $\zeta^2$ can be large for some FL problems when the clients naturally have highly heterogeneous data. In such situations, the lower bound is really pessimistic. Although this fact (Theorem III) is expected for the experts in optimization, it is very important for the field of Byzantine-robust optimization: it creates a clear picture of the limits of robustness.

2. **New upper bounds for Byzantine-robust optimization under bounded heterogeneity.** The authors prove new complexity bounds for Byzantine-robust optimization under bounded heterogeneity in the non-convex case. The derived results match the established lower bound. Moreover, it is shown that one can achieve any predefined accuracy (Theorem IV) when the heterogeneity level at the point $x$ is proportional to $\|\nabla f(x)\|^2$ (can be seen as a strong growth condition from (Vaswani et al., 2019)).

3. **Bucketing as a tool to make Krum, Geometric Median, and Coordinate-wise Median robust.** (Karimireddy et al., 2021) show that even in the homogeneous case Krum, Geometric Median, and  Coordinate-wise Median are provably non-robust to Byzantine attacks. This work fixes this drawback of the mentioned aggregation rules (Theorem I) via a simple tool called bucketing. This result (together with Theorem II) is very important for Byzantine-tolerant optimization in the homogeneous case, since Krum, Geometric Median, and Coordinate-wise Median were not analyzed previously without restrictive assumptions.

4. **Clarity.** The paper is well-motivated, clearly written, and has a good structure.

## Weaknesses
1. **Inaccuracies in the proofs.** I have checked all the proofs and noticed several inaccuracies and unexplained derivations. Although it is possible to fix all the issues, in the current shape, it is hard to follow some parts of the proofs. I list all my questions and comments in section **Questions and comments about the proofs** of my review.

2. **Comparison with related work.** Unfortunately, it is not trivial to compare the results from this paper with other related works given the information provided in the related work section. The current version of the related work section summarizes the known works without going into the details. However, it is important for the paper to provide an explicit comparison with the related works (with the discussion of the derived rates and assumptions). I strongly encourage the authors to provide such a comparison at least in the appendix. Moreover, the authors should pay a lot of attention to the comparison with Yang & Li (2021) since they also use bucketing.

## General questions and comments
1. **Abstract, sentence "Our work is the first...":** This sentence is not correct since Li et al. (2019) also derive convergence results under realistic assumptions for strongly convex problems. Perhaps, the authors wanted to emphasize that their work provides the first guarantees under the not too strong assumption in the non-convex case.

2. **Page 2, "However, none of these methods are applicable to the standard federated learning."** This claim requires additional clarifications. For example, it is not clear why the method from Li et al. (2019) is not applicable.

3. **Definition A:** I suggest the authors additionally emphasize that this definition is useful for both homogeneous and heterogeneous cases.

4. **Remark 3:** This is 1 iteration of CClip, which is not necessarily the output of the aggregator.

5. **Second paragraph after Remark 4, "... which matches the optimal iid Byzantine robust rates of (Karimireddy et al., 2021)".** It is not clear why the mentioned rate is optimal (and in what setting). Karimireddy et al. (2021) do not prove the lower bound. Moreover, there is a recent work, where the better rate is achieved (in terms of the number of iterations): Gorbunov et al. "Secure Distributed Training at Scale." arXiv preprint arXiv:2106.11257 (2021).

6. **Page 7, the sentence above Theorem 7, "... typically holds in most realistic settings (Vaswani et al., 2019".** Vaswani et al. (2019) do not provide evidence that this assumption holds in most realistic settings. In fact, they show it (Strong Growth Condition) for an example of squared hinge-loss in the case of linearly separable data. They also prove that Strong Growth Condition follows from the interpolation condition when the summands in the loss function are smooth and the loss satisfies PL-condition. However, in this case, the known bound upper bound for $B$ is proportional to $\frac{L}{\mu}$ where $L$ is the maximal smoothness constant of $f_i$ and $\mu$ is the PL-parameter of $f$. Therefore, the current theoretical estimates for $B$ are quite large even for simple special cases.

7. **Theorem IV, condition $B^2 < \frac{1}{3c\delta}$.** In view of my previous comment, this requirement may imply that $\delta$ is tiny. I think the discussion of this requirement should be added to the paper.

8. **Missing reference.** This work also addresses the heterogeneous case for Byzantine-robust optimization: *Zhaoxian Wu, Qing Ling, Tianyi Chen, and Georgios B Giannakis. Federated variance-reduced stochastic gradient descent with robustness to byzantine attacks. IEEE Transactions on Signal Processing, 68:4583–4596, 2020.* Moreover, Wu et al. (2020) also prove convergence guarantees under similar assumptions. Therefore, a detailed comparison of the derived results should be added.

9. **Conclusion, the last sentence, "... our results represent a major breakthrough..."** Although the work makes a strong contribution to the field, in my opinion, it cannot be called a breakthrough taking into account that many of the building blocks were known and analyzed to some extent (bucketing, client momentum). I think only a couple of papers can be called a breakthrough objectively (e.g., Nesterov's acceleration and discoveries of the same caliber). So, I suggest the authors rewrite the sentence: let the readers decide for themselves whether this paper is a breakthrough or not.

## Questions and comments about the proofs
1. **Proof of Lemma 1, formula for $\mathbb{E}_{\pi}[y_i | i \in \widetilde{\mathcal{G}}]$:** this is true, but the detailed derivation should be added.

2. **Lemma 7.** First of all, the lower and upper bounds should be multiplied by $n$. Moreover, I have the following question about the proof of the lower bound: does the described distribution of $y_i$ correspond to any distribution of the initial vectors $x_i$ before the bucketing? This is crucial for the correctness of the lower bound.

3. **Page 20, robustness of Krum.** The first formula is not proven. Moreover, it is not clear what is $S^{\ast}$. Next, in the formula above "Taking expectation now on both sides yields" the minimization should be taken over the sets $S$ such that $|S| = \frac{3m}{4}$. In the next formula, it seems that the numerator should contain $4n\tilde{\rho}^2$. After that, the sentence "Then, the number of Byzantine workers can be bounded as $|\tilde{\mathcal{B}}| \leq m(1/4 - \delta)$" should be rewritten as "Then, the number of Byzantine buckets can be bounded as $|\tilde{\mathcal{B}}| \leq m(1/4 - \nu)$". Finally, in the upper bound for $\mathbb{E}\|y_{k^\ast} - \bar{x}\|^2$ the denominator of the first fraction should have $\nu m$ instead of $\nu n$.

4. **Page 21, robustness of Geometric median.** In the third sentence, the word "worker" should be replaced by "set"/"bucket". There is also a typo in the formula after the words "Squaring both sides, expanding, ...": the sum in the first row should not have $\mathbb{E}$ inside.

5. **Appendix D, the proof of Theorem III.** The first formula should contain $\delta$ instead of $\hat\delta$. Next, the formula after the words "Note that the gradient heterogeneity ..." should be supported by the full derivation (or explained). It is true but requires few extra steps.

6. **Proof of Lemma 8** contains several inaccuracies and unexplained derivations. First of all, what is $\mathbb{E}[\cdot | i]$? If it is an expectation conditioned on $i$, then $\mathbb{E}[g_i(x^{t-1}) | i] \neq \nabla f_i(x^{t-1})$ since $x^{t-1}$ depends on the stochasticity not related to the choice of $i$. But the proof uses $\mathbb{E}[g_i(x^{t-1}) | i] = \nabla f_i(x^{t-1})$. This should be fixed (here and below). Next, it is not clear what is $\mathbb{E}_i[\cdot]$. The first formula on page 24 is incorrect due to the same reason as the first formula in the proof. This issue should be fixed as well. The second formula on page 24 is also inaccurate: the RHS should have $\zeta^2(1 - (1-\alpha)^t)$. Next, there is a typo in the sentence "This is because the randomness in the sampling...". Finally, the last sentence in the proof the authors claim that it is enough to apply Definition A, but it is not correct: this definition requires $\mathbb{E}\|m_i^t - m_j^t\|^2 \leq \rho_t^2$ while the authors prove a weaker result that $\mathbb{E}_i\|m_i^t - \bar{m}^t\|^2 \leq \rho_t^2$. Overall, the proof of Lemma 8 requires a major revision (and, probably, Definition A should also be changed to fit the proof).

7. **Lemma 10.** Numerical constants are incorrect: instead of $\frac{2\alpha}{5}$ the formula should have $\frac{5\alpha}{16}$ and instead of $\frac{\alpha}{10}$ one should have $\frac{3\alpha}{32}$.

8. **The proof of Theorem V** contains several places that should be better explained (and, most likely, corrected, because of the inaccuracies). First of all, it seems that the authors forgot to upper bound $\mathbb{E}\|m_t - \bar{m}_t\|^2$: it is contained in the RHS of the second formula in the proof, but it is omitted in the derivations on page 26. Next, the last formula on page 25 is inaccurate: one should have $\nabla f(x^{t-1})$ in the RHS. Moreover, the next step in the proof is unclear: it seems that a lot of derivations are omitted. For me, it is not clear how the term $\mathbb{E}\|m_t - \bar{m}_t\|^2$ is handled. Therefore, this derivation should be significantly rewritten and checked.

## Minor comments
1. **$|\mathcal{B}| = f$.** The symbol $f$ is already used to denote the objective function. I suggest using a different notation for the number of Byzantines.

2. **Lemma 1.** $\widetilde{\mathcal{G}}$ is defined only in the proof. The authors should add the definition in the statement of the lemma.

3. **Figure 9** contains low-resolution images. The authors should replace them with the ones with higher resolution.

4. **Page 18, the last formula:** full stop is missing in the end of the of the formula (please, check other formulas that end sentences as well).

5. **Page 19, the second sentence, "... each can belong to only 1 bucket each":** one "each" should be removed.

6. **Lemma 9.** In this lemma, the authors use $m_t$ instead of $m^t$. The notation should be unified.

7. **Lemma 10.** The authors use $x_{t-2}$ and $x^{t-1}$. The position of the indices should be unified.

**Summary Of The Paper:**

The paper provides a systematic and deep theoretical study of the problem of Byzantine-robustness in the heterogeneous setup, i.e., when workers have non-identical datasets and, as s result, their local loss functions are non-identical. The authors prove that even under bounded heterogeneity it is impossible to provably achieve any predefined accuracy of the solution by any method. Next, the authors propose and analyze an algorithmic tool called *bucketing* and show that it makes some known aggregators such as Krum, coordinate-wise median, and geometric median to be *agnostic* robust aggregators. Under bounded heterogeneity assumption, the authors derive that Robust Client Momentum method from (Karimireddy et al., 2021) converges for non-convex problems to a neighborhood of a stationary point. Next, it is shown that the size of the neighborhood can be made arbitrarily small for over-parameterized problems. Numerical experiments corroborate theoretical findings and, in particular, show the benefits of the proposed bucketing procedure.

Overall, the paper is well-motivated, clearly written, and contains solid contributions. There are also some minor inaccuracies in the proofs, small typos, and other minor weaknesses -- I list them below. I encourage the authors to address all of them.

**Summary Of The Review:**

To sum up, I am sure that the paper should be accepted to the conference after minor improvements. If the authors apply all necessary corrections and address my comments properly, I will increase my score: the paper deserves acceptance as a spotlight or even oral talk.

---

> ### Author Response · Authors · 2021-11-22
> **Reply to Reviewer ago1**
>
> We thank the review ago1 for the great effort spent on reviewing and proofreading the paper and valuable questions and suggestions. We have included all of the minor comments into the paper and added additional discussion on the related works in appendix F.1.
>
> ### Reply to“General Questions and Comments”
>
> - We agree with bullet points 1, 5, 9 and we have tone down the sentence in the paper.
> - Bullet point 2: RSA can be viewed as a variant of federated learning. We have rearranged the related works to address this issue.
> - Bullet point 3 & 4: We have incorporated it in the paper.
> - Reply to bullet point 6 & 7 on B^2:
>
>     Using terminology of Vaswani et al. 2019, the strong growth condition has been extensively used in optimization for deep learning (Vaswani et al., 2019; Ment et al., 2020; Vaswani et al., 2020). This line of work is summarized in [this excellent talk](https://www.youtube.com/watch?v=nk4M-kYvaNU), which shows how the strong growth assumption is both realistic and (perhaps more importantly) useful in understanding optimization algorithms in deep learning. We also empirically show that it holds in our setting in Figure 7 in the appendix.
>
>      Having said this, there are two limitations to our work. Firstly, the strong growth condition is an attempt to formalize the intuition that our deep learning model has the right inductive bias and hence fits the correct data easier than incorrect data. However, as the reviewer notes, it currently lacks any theoretical justification and more foundational work is necessary. Secondly, we cannot prove that the strong growth condition is required for Byzantine robustness (only that it is sufficient). In fact, we believe that our analysis can be strengthened in the convex case to prove that simple overparameterization (i.e. weak growth) is sufficient. This latter condition is also necessary [(Gupta et al. 2020)[https://arxiv.org/abs/2003.09675]].
>
>     Bullet point 7: we have added a comment after Theorem V.
>
>
> - Reply to bullet point 8 on comparison with Byrd SAGA:
>
>     Thank you for the pointer to this highly relevant paper. In our setting, there are two sources of variances of the gradients - intra-worker variance $\sigma^2$ and inter-client variance $\zeta^2$. We show that simply using worker momentum suffices to tackle the former and handling the latter $\zeta^2$ is the main challenge. ByrdSAGA assumes that each worker only has finite data points (as opposed to the stochastic setting we consider). Hence they can use SAGA on the worker in place of worker momentum to reduce the intra-client variance $\sigma^2$. The effect of $\zeta^2$ (which is our main focus) remains unaffected.
>
>     Further, they consider the strongly convex setting whereas we analyze non-convex functions. Ignoring $\mu$ for sake of comparison, their Theorem 1 proves convergence to a radius of $\Delta_1 = O(\zeta^2)$ since always $C_\alpha \geq 2$. Thus, their rates are similar to (Acharya et al. 2021) and do not converge to the optimum even when $\delta = 0$. In contrast, our Theorem II proves convergence to a radius of $O(\delta \zeta^2)$. We believe our improved handling of $\zeta^2$ can be combined with their usage of SAGA/variance reduction to yield even faster rates. This we leave for future work.
>
>
>
>
> ### Reply to “Questions and comments about the proofs”.
>
> - We have fixed the issues/typos in bullet point 1, 2 and 4.
>
>     Reply to 2 (Lemma 7): Indeed our lower bound is for a particular distribution of $y_i$. But, Lemma 7 is only used to illustrate the mistake in previous proof of Krum by [Blanchard et al. 2017]. Our proof does not rely on it. Having said this, a similar bound holds (albeit with more computations) if we instead start with $x_i \sim \rho\sqrt{n}\text{Bern}(p = \tfrac{1}{n})$.
>
> - About bullet point 7, the coefficients in lemma 10 (e.g. ⅖, 1/10) are correct but a bit looser than 5/16 and 3/32. We did this for readability.
> - Bullet point 3: The first formula is an alternative form of the krum --- we replace the original minimizing over the sum of closest distance with a min-max formulation. We have corrected other constants.
> - Bullet point 6: The main confusion for the readers of Lemma 8 comes from the use of two expectations: one for $\sigma^2$ and one for $\zeta^2$. Now we have updated the proof by explicitly distinguishing the two expectations while keeping the proof structure unchanged. This can address the first part of bullet point 6. We updated the rest accordingly.
>
> - Bullet point 8: The $\mathbb{E} \| m^t - \bar{m}^t \|^2$ appear together with $\mathbb{E} \| m^{t-1} - \bar{m}^{t-1} \|^2$ in the  first equation of the proof. We bounded the $\mathbb{E} \| m^{t-1} - \bar{m}^{t-1} \|^2$ using Lemma 8 but forgot to bound $\mathbb{E} \| m^{t} - \bar{m}^{t} \|^2$. Thus we handle $\mathbb{E} \| m^{t} - \bar{m}^{t} \|^2$ similarly with Lemma 8 and updated rest of proof proceeds similarly  to the original one except for some changes of constants.

---

> > ### Comment · Reviewer_ago1 · 2021-11-26
> > **Thanks for the reply! I am increasing my score**
> >
> > I thank the authors for addressing all my concerns and modifying the paper accordingly. I have read other reviews and responses, the main criticism was properly addressed by the authors. Therefore, as I promised, I have increased my score to 10 -- the paper deserves this.
> >
> > I also have several minor comments about the revised version.
> >
> > 1. In Remark 3, the text almost overlaps with the formula. This should be fixed in the final version.
> >
> > 2. Could the authors provide formal proof of why the min-max formulation of Krum is equivalent to the one from the paper? It is not straightforward to see why this is true.

---

> > > ### Author Response · Authors · 2021-11-28
> > > **Reply to Reviewer ago1**
> > >
> > > We thank Reviewer ago1 again for the giving us higher scores as well as the suggestions. We will update the remark 3 in the later version. We agree that min-max formulation of Krum is not straightforward and actually not used in the proof, so we will just use the original definition of Krum.

---

### Official Review · Reviewer_EczS · 2021-11-02

**Correctness:** 3
**Technical Novelty And Significance:** 3
**Empirical Novelty And Significance:** 3
**Recommendation:** 8
**Confidence:** 4

**Main Review:**

## Strengths
The bucketing scheme proposed in this paper is interesting. It is easy to adopt and does not bring much extra computational cost, but can improve the performance of existing Byzantine-resilient algorithms, as shown in the empirical results. Also, I appreciate that the authors provide convergence results for general aggregation rules, together with precise analysis for three common aggregation rules in Theorem I.

## Weaknesses
When decreasing the heterogeneity, the bucketing scheme also reduces the number of candidate vectors (gradients). However, in the worst case, the number of corrupted vectors will remain the same. Thus, the number of Byzantine workers that can be tolerated will decrease to $1/s$ when adopting bucketing. Although the empirical results show that $s=2$ is sufficient to overcome heterogeneity, the number of Byzantine workers that can be tolerated already drops by half in this case. I am not aiming to criticize the bucketing scheme but to point out the limitation. I think that this problem has truly restricted the application prospects of bucketing in general cases.


**Summary Of The Paper:**

This paper proposes a bucketing scheme for robust distributed learning with heterogeneous data. Gradients are more homogeneous after bucketing, which will increase the robustness of existing algorithms. The author provides theoretical analysis as well as empirical results.

**Summary Of The Review:**

Although there are weaknesses in this work, considering the challenges from heterogeneity, I think this work is slightly above the threshold.

--------
(Post-rebuttal)
My major concerns have been properly addressed by the authors and the quality of this paper has been improved after revision. Thus, I have raised my rating.

---

> ### Author Response · Authors · 2021-11-22
> **Reply to Reviewer EczS**
>
> We thank Reviewer EczS for the kind review. We agree that it would be best to keep maximum tolerated workers to 50%. We note that this limitation is not uncommon, e.g. aggregator Bulyan in this well cited paper (El Mhamdi et al., 2018) also tolerates at most 25% of Byzantine workers. In many practical applications like federated learning, it still makes sense to assume that the adversary can not control over 25% of the participants - we expect perhaps at most 5% of workers to be malicious in the real world.
>
> El Mhamdi M, Guerraoui R, Rouault S. The hidden vulnerability of distributed learning in byzantium.  International Conference on Machine Learning. PMLR, 2018: 3521-3530.

---

> > ### Comment · Reviewer_EczS · 2021-11-29
> > **Inclined to raise my rating, but some concerns remain**
> >
> > I greatly thank the authors for the explanation, but there are still some concerns as listed below.
> >
> > 1. **About reduced maximum tolerated workers.** I agree with the authors that the fraction of Byzantine clients is usually small in real-world applications, but it is necessary to make the readers aware of the limitation.
> >
> > 2. **About comparison with (Yang & Li, 2021).** I appreciate that the authors added a comparison with existing works in appendix F. I agree with the authors that the motivation of their method is vastly different from BASGD. However, the statement *"our theoretical analysis does not require bounded gradient assumption ||∇f (x)|| ≤ D for all x."* seems improper. The bounded gradient assumption is usually used to bound the error caused by asynchrony, which is not a problem in this submitted work since it is a synchronous method. Therefore, the comparison seems unfair.
> >
> > 3. (minor) **About the references.** The authors could be more careful about the format of the references. Specifically, letters in some words should be capitalized such as 'SGD' and 'Byzantine'. Besides, the format of some references is not consistent.
> >
> > Overall, the quality of this paper has been improved after revision. I am inclined to raise my rating if the authors could properly address my concerns.

---

> > > ### Author Response · Authors · 2021-11-29
> > > **Reply to update**
> > >
> > > We thank the reviewer for the suggestions, and appreciate their willingness to revise the score.
> > >
> > > We will be sure to highlight the limitations of our work w.r.t. number of Byzantine workers tolerated, and also clean up our references in our next version (unfortunately we are no longer able to directly update the submission).
> > >
> > > We also agree that the bounded gradient assumption is common in asynchronous analyses and will remove this objection. Instead, we will focus on the difference in methodology and motivations.

---

> > > > ### Comment · Reviewer_EczS · 2021-11-29
> > > > **Reply**
> > > >
> > > > I appreciate the authors for their reply and the effort in modifying the paper. Therefore, I have raised my rating accordingly.

---

### Official Review · Reviewer_WV2X · 2021-11-02

**Correctness:** 4
**Technical Novelty And Significance:** 3
**Empirical Novelty And Significance:** 3
**Recommendation:** 8
**Confidence:** 3

**Main Review:**

Strengths:
-I think that this is an elegant way of combining two key areas of work in the FL community: non-iid data sets + Byzantine nodes
-Empirical results seem robust
-The bucketing model is simple, believable, and it is nice that it composes well with existing methods.
-The theoretical analysis of the model is extensive: it matches existing bounds when \delta or \rho=0, and the lower bound matches the non-trivial term that occurs when both \delta,\rho \neq 0.

Weaknesses:
-Are there more complicated examples of non-iid models under non-adversarial attacks that perform poorly? The Rademacher distribution is a good starting point to justify the analysis, but seeing the performance of existing aggregators on more complicated data sets when \delta=0 would be interesting.
- Still not convinced about the upper bound on gradient norm in the over-parameterized setting (where the term is bounded by B \times \grad f(x)). Is this realistic? Perhaps I am not seeing how this falls out of over-parameterization. A note on this might be useful in the write-up of the paper.
-The lower bound analysis (Theorem 3) could have some more intuition as well, in terms of what the argument is like, and what the functions that are used consist of. In particular, it would be interesting to note whether the author thinks the adversarial functions for the bound are efficiently computable, and if not, whether hard functions can occur in practice over the byzantine nodes (could this happen without some amount of coordination amongst the byzantine nodes for example?)
-What about other types of attack objectives, i.e. backdoor attacks (this seems highly relevant in the non-iid setting, especially if the adversary has knowledge of which clients have which data)


**Summary Of The Paper:**

This paper focuses on Byzantine-robust federated learning, a learning paradigm where a centralized server coordinates learning across a data set partitioned across multiple worker nodes, some of which are adversarial. Typically this is done via robust aggregation schemes which ensure that adversarial nodes do not hinder learning.  In this setting, the author's main focus is on studying the setting where honest worker nodes own heterogeneous data sets. As the authors mention, the existing literature on Byzantine-robust federated learning focuses on the setting in which honest worker nodes draw iid samples from an underlying data distribution, and although there is also existing literature on (non-Byzantine) federated learning over heterogenous data distributions amongst nodes, this work brings these two strands of research together.

The main results are the following:
-Providing (simple) settings and empirical results wherein common aggregation methods fail in the presence of heterogeneous data, even when there are no adversarial nodes.
-Providing a specific attack vector for the heterogeneous Byzantine setting, whereby adversarial nodes choose an honest node to replicate ("Mimic" attack). This exposition comes with an efficient algorithm for computing the most "hurtful" choice of node to mimic, and the authors empirically demonstrate the potential of this attack against common aggregation methods
-Formally defining the design objective of "Agnostic Robust Aggregators", which quantifies degradation of aggregator performance as a function of byzantine tolerance and dataset heterogeneity.
-Providing a simple randomized bucketing scheme. Messages from nodes are randomly averaged in buckets, where the number of buckets is a free parameter. Increasing the number of buckets reduces the variance of bucket representatives, but potentially increases the number of byzantine agents (post-bucketing). If there is sufficient margin in existing aggregation protocols, performing judicious bucketing before applying the aggregation rule permits making Agnostic Robust Aggregators out of existing aggregation schemes (with specifically quantifiable guarantees).
-The authors also theoretically study optimization methods that make use of aggregation methods (via stochastic gradient descent). In this setting they provide upper bounds on convergence rates of optimization using robust aggregators (in terms of the parameters that govern the robustness of aggregation method as per Definition A), and show that these upper bounds match existing upper bounds in the case where there are no byzantine agents (\delta = 0) or when data is homogeneous (\rho = 0). In the regime where these terms are non-zero, convergence is not guaranteed, but via information theoretic methods, the authors also provide a matching lower bound
-If heterogeneity bounds are more refined (whereby gradient variation is bounded by the order global gradient of the entire dataset), then the authors demonstrate convergence of robust aggregation methods on SGD. This setting applies when systems are over-parameterized.
-Finally, the authors provide multiple experimental results that validate the theoretical findings of the work.

**Summary Of The Review:**

I recommend this paper for acceptance. This seems like a fitting extension of existing work, and the authors have provided a corresponding framework for quantifying performance loss in the presence of byzantine agents and heterogeneous data for federated learning. Their results match existing work in regimes when there are no byzantine agents, and when there is a lack of heterogenous data. Furthermore, the non-trivial loss of performance in the regime where both byzantine agents and heterogenous data is present is accounted for in a matching lower bound. These results will be of interest to the growing body of work in federated learning at ICLR.

---

> ### Author Response · Authors · 2021-11-22
> **Reply to Reviewer WV2X**
>
> We thank Reviewer WV2X for the detailed summary, insights and appreciation of our paper. We address the weakness as follows.
>
> - Table 1 and 3 indeed show more **complex data** settings (MNIST) with no attackers. The performances of median based aggregators also drop by around 10% on non-IID data compared to the IID case here as well.
>
> - This assumption is the “**strong growth condition**” (Schmidt & Roux, 2013) and it has been studied in many past works (Vaswani et al., 2019; Ment et al., 2020; Vaswani et al., 2020). We empirically evaluated the value of $B^2$ in Figure 7 in the appendix. The conclusions from this line of work has been summarized in [this talk](https://www.youtube.com/watch?v=nk4M-kYvaNU) by Mark Schmidt. It discusses how the strong growth assumption is both realistic and useful in understanding the performance of optimization algorithms in deep learning.
>
> - **Intuition behind lower bound**. For simplicity, consider the following centralized setting which illustrates the key difficulty. Suppose that we are given some training data, **upto** 10% of which may be poisoned. We train on this data using whatever algorithm we want, and at the end we get a train accuracy of 90%. Are we not able to fit the remaining 10% data because it is the poisoned part, or is it because it was more difficult than the rest? If it is the former, we should leave the 10% as is. If it is the latter and there wasn’t any poisoned data, we should perhaps use a bigger model and try to fit the entire data. However, without being given additional information there is no way to know which of the scenarios we are in.
>
> 	The lower bound in the federated setting works exactly as above. Some of the workers may be malicious, or may simply have difficult/weird data. Without additional information, there is no way to tell which. Note also that the above argument does not need any coordination - just that we do not completely fit all the data. Hence, we cannot hope to prove stronger guarantees if we assume no coordination.
>
> - **Backdoor attacks** are very interesting, but their goal is to lower the accuracy on any subtask while keeping the accuracy on main tasks. This is different from our setting where attackers aim to lower the accuracy on the main tasks. Therefore defending backdoor attacks requires optimizing over the worst subgroup of the objectives and we need a different theoretical framework. We believe that by combining our Byzantine robust algorithms with techniques from subgroup fairness/robustness literature could defend against backdoor attacks. We leave this for future work.

---

> > ### Comment · Reviewer_WV2X · 2021-11-29
> > **Response to author reply**
> >
> > I appreciate the authors' response to the review. With regards to the intuition regarding the lower bound, I was thinking more along the lines of what classes of functions are used / whether implementing such an attack is computationally intensive, however upon taking a further glance at the supplementary material I see that the specific class of functions/ argument is straightforward. Otherwise, all points made by the authors in the reply are satisfactory and my score remains the same.

---

### Official Review · Reviewer_oF9Y · 2021-11-03

**Correctness:** 3
**Technical Novelty And Significance:** 3
**Empirical Novelty And Significance:** 3
**Recommendation:** 6
**Confidence:** 4

**Main Review:**

0. The investigated problem, Byzantine-robust distributed learning over non-iid data, is important. The algorithm development and the analysis both contain new results
1. In the previous version of this paper, the authors propose resampling to reduce the heterogeneity across the workers. Why do the authors switch to bucketing in this version?
2. In Page 2, the authors claim that “none of these (non-iid Byzantine-robust) methods are applicable to the standard federated learning.” Please justify this claim.
3. Related to the above comment, the authors do not compare the proposed methods with any other non-iid Byzantine-robust methods.
4. Also about the numerical experiments, the MNIST dataset is too simple. Testing on one or two additional large datasets could make the results more convincing (in particular, for the over-parametrized case).
5. In Section 3.2, the mentioned mimic attack has appeared in other papers that investigate non-iid Byzantine-robustness. Please cite.
6. In Section 4.2, why place centered clipping here? It should be located in Section 3.
7. The role of momentum is not well investigated. The authors should indicate its contribution to the performance improvement.
8. For the numerical experiments of the over-parametrized case, the authors use centered clipping as the basic aggregator. But centered clipping has certain robustness in the non-iid setting. Using other base aggregators, such as geometric median, could be better here.
9. Theorem IV requires that B^2 is sufficiently small. Please check whether this condition can be satisfied in the numerical experiments.


**Summary Of The Paper:**

In this paper, the authors consider Byzantine-robustness of a distributed learning system in the setting of non-iid data distribution. The bucketing technique is applied to reduce the heterogeneity across the workers. To further reduce the variance within each worker, the authors adopts the momentum technique. These two techniques can be combined with various robust aggregation rules. The authors prove the convergence of the combined methods, and show that they reach the lower bound. The authors also claim that for the over-parametrized case, the negative impact of data heterogeneity can be eliminated.

**Summary Of The Review:**

Overall, this paper has publication merits, but some issues need to be addressed.

---

> ### Author Response · Authors · 2021-11-22
> **Reply to Reviewer oF9Y**
>
> We thank the reviewer oF9Y for the valuable suggestions and insights. We address the questions as follows:
>
> 1. We found that the analysis and performance of bucketing and resampling are essentially the same. However, bucketing is conceptually simpler and computationally more efficient as a result of no gradient replication. Thus we chose to replace resampling with bucketing. In Appendix A.2.4 we provided an empirical comparison between them which supports that these two methods have very similar accuracy.
>
> 2. > “However, none of these methods are applicable to the standard federated learning.”
>
>     As we note in the related work section, some of these works focus on data center setting or volunteer training where multiple workers can be assigned with the same data by the server. Others assume that the server in federated learning has a clean dataset which can be leveraged to evaluate the worker gradients. However, in a standard federated learning setup there is no training data stored on the server, thus these methods could not be applied. (see also 3. below)
>
> 3. The closest work on Byzantine robust learning with heterogeneous data is (Data & Diggavi, 2021) which is a theoretical work and there is no empirical evaluation in the paper. Further, the spectral filtering method used there is computationally expensive. Thus we didn’t compare with their method. The work of (Li et al., 2019) assumes the strong convexity of the loss function and in addition requires an L1 regularizer, both of which are not applicable to the deep learning setting we consider. So we didn’t include them in the comparison.
>
> 4. We will add more experiments on CIFAR-10 to the appendix when the experiments finish. We note that given the simplicity of MNIST, the drop in performance of the baseline methods is even more remarkable. This highlights the need for our work.
>
> 5. Sorry that we are not aware of the mimic attacks in other papers. We would be happy to include any references the reviewer may point out.
>
> 6. This is a good point. Our initial motivation is that the CM/Krum/RFA are all median based methods and they all suffer from heterogeneity in a similar way. We found it simpler to explain these 3 methods together and introduce CClip separately.
>
> 7. We demonstrated the effect of momentum at the end of page 8 and beginning of page 9. In Figure 1, we use dashed lines to indicate training with momentum ($\beta=0.9$). The main conclusion we drew for momentum is at the end of page 8. “In the last two columns of Figure 1 we show that worker momentum and bucketing reduce such variance while momentum alone is not enough.”. That is, the momentum reduces the variance within a client but does not reduce the heterogeneity among workers which could be improved together with bucketing.
>
> 8. We are indeed already using RFA for the overparameterized experiments. We have updated the experiment description to highlight this. This can also be verified by our anonymous code submission here: https://anonymous.4open.science/r/byzantine-robust-noniid-optimizer-F58F/exp8.py
>
> 9. In the appendix A.2.3 we empirically measure $B^2$ and observe that $B^2$ decreases with increasing the degree of overparameterization. So the condition holds true if the model is overparameterized enough or if the $\delta$ is small. But even if this condition does not hold true, it does not mean the iterates will diverge. In the appendix E, we present a more general Theorem V which captures the heterogeneity with $\zeta^2$ and $B^2$ simultaneously, and show that the iterates eventually converge to a neighborhood of the optimum even if $\zeta^2 > 0$. We have added a note after the Theorem V.
>
>
> Data D, Diggavi S. Byzantine-resilient SGD in high dimensions on heterogeneous data[C]//2021 IEEE International Symposium on Information Theory (ISIT). IEEE, 2021: 2310-2315.

---

> > ### Comment · Reviewer_oF9Y · 2021-11-27
> > **Updates**
> >
> > I thank the authors for the replies. In my opinion, the authors still need to revise several places, avoid over-claiming, and correct theoretical mistakes.
> > 1. Fair comparison with other works. This is related to my previous comments 2 and 3, as well as several comments from the other reviewers. At least, the authors can compare the proposed algorithm with that in (Li et al 2019). Although it assumes strongly convex loss function and adds an L1 regularization term, the algorithm itself is still applicable to training a deep neural network.
> > 2. Mimic attack. This is related to my previous comment 5. It is exactly the attack on the heterogeneous data in (Li et al 2019). Please check if it has appeared earlier, cite proper papers, and avoid emphasizing “we propose a new attack”.
> > 3. Recovering the SGD rate when \delta=0. A strong claim, which appears several times in this paper, is that the developed algorithms can recover the SGD rate and achieve exact convergence. However, the analysis is wrong and the claim is incorrect (or at least, very improper). See details below.
> > - In Theorem 1, the authors require that s=\delta_\max/\delta, which is \infty when \delta=0. If we understand from the resampling perspective, it means that all stochastic gradients (now they are all correct) are duplicated for infinite times, resampled, averaged within each bucket, and sent for robust aggregation. In this case, the averages of the buckets are the same, and the algorithm is exactly SGD. However, this algorithm is impractical since we cannot afford infinite duplications. If we set s as a finite number, then exact convergence is no longer possible.
> > - Although we can use resampling to explain the case of \delta=0 (and s=\infty), bucketing is not applicable here – Line 3 of Algorithm 1 cannot work.
> > - Theoretically, \delta=0 affects Definition A, because |\mathcal{G}>(1-\delta)n| is impossible. Perhaps this one can be fixed. But it also affects the proof of Theorem 1 – to fix it, the authors should either exclude \delta=0, or introduce infinite duplications.

---

> > > ### Author Response · Authors · 2021-11-28
> > > **Comparison with [Li et al. 2019] and the $\delta=0$ corner case**
> > >
> > > We thank the reviewer for the suggestions.
> > >
> > > - Bullet point 1： There exist very simple settings where the *RSA algorithm does not converge* without regularization. Suppose $f_0 =0$, we take only 1 local step ($K=1$), and only 1 worker ($n=1$) without any Byzantine attackers. RSA becomes equivalent to SignSGD, which does not converge [Karimireddy et al. 2019]. Changing the value of $p$ wouldn’t help either since there exist 1D counterexamples. Having said this, we would be happy to add some experimental comparisons with RSA.
> > >
> > > - Bullet point 2：We thank the reviewer for this reference. We missed this part as this is not explicitly phrased an attack. We will ensure due credit is given to (Li et al. 2019).
> > >
> > > - > 3.1 $s=\delta_\max/\delta$, which is $\infty$ when $\delta=0$
> > >
> > >     The theoretical results for the reduction in variance due to bucketing were simplified for the ease of presentation, and the infinity is merely an artifact of this.
> > >
> > >     Clearly, *when $s=n$, we already use only 1 bucket and there is no variance*. Let us now see this slightly more tighter version which utilizes the fact that bucketing samples without replacement. In Lemma 1, the new variance after the Bucketing procedure can be tightened to $\frac{\rho^2(n-s)}{s^2}$. Using this in Theorem I, we need to pick the smallest $s$ such that $\frac{(n-s)}{s^2} \leq \frac{\delta}{\delta_{\max}}$. Clearly, this always yields a finite value of $s \leq n$ and $s=n$ when $\delta = 0$ as required.
> > > We thank the reviewer for pointing out this potential confusion and will clarify in our paper as well.
> > >
> > > - > 3.2 Bucketing is not applicable when $\delta=0$here. Line 3 of Algorithm 1 cannot work
> > >
> > >     Unlike resampling, bucketing still applies in this case. Even if $ s > n$, we would use $\lceil \frac{n}{s} \rceil = 1$ bucket. Anyways, as we explain above, we never need to choose $s > n$ since $s=n$ already makes the heterogeneity 0 (everyone is in the same bucket).
> > >
> > > - > 3.3 $ \mathcal{G}>(1-\delta)n $ is not possible when $\delta = 0$.
> > >
> > >     This was a typo and should have been $ \mathcal{G} \geq (1-\delta)n$. This is correctly written elsewhere (e.g. statements of Lemma 1, Theorem I, etc.). All proofs use the correct definition. Thank you for the pointer.

---

> > > > ### Comment · Reviewer_oF9Y · 2021-11-28
> > > > **Updates**
> > > >
> > > > I thank the authors for the clarifications. For comment 3, please modify the Theorems and proofs correspondingly.
> > > >
> > > > I also strongly recommend the authors to reconsider the necessity of emphasizing “recovering the SGD rate when \delta=0”. It is possible because the proposed algorithm sets s=n, uses one bucket, and applies mean aggregation within the bucket. But if any malicious workers may exist, we will never run such a risk. On the other hand, when there are no malicious workers, one can always switch from any other Byzantine-robust to SGD.

---

> > > > > ### Author Response · Authors · 2021-11-29
> > > > > **Interpreting $\delta$ and importance of $\delta=0$**
> > > > >
> > > > > We view $\delta$ as a knob which can be used by the practictioner to encode their confidence. If the practioner is very confident that there aren't likely to be too many attackers (e.g. due to contractual agreements, ToS, etc.) they can set this value to be small (say 0.01--0.05). On the other hand, if they believe there are going to be a large number of possible attackers, it can be set larger (say 0.2--0.25).
> > > > >
> > > > > The tradeoff one hopes to make is that allowing for smaller values of $\delta$ can yield faster convergence. However, we want to be sure that we make this tradeoff as well as possible (Pareto optimal). While we don't have proof of Pareto optimality, the fact we recover ideal SGD rate when $\delta$ is 0 should be seen as evidence that our algorithms do a reasonable job trading off robustness against efficiency.

---

> > > > > > ### Comment · Reviewer_oF9Y · 2021-11-30
> > > > > > **Updates**
> > > > > >
> > > > > > In fact, I like the \delta-s tradeoff since it reveals the importance of reducing heterogeneity to Byzantine-robustness (even though the authors set s as a small value in the numerical experiments). The case of \delta=0 and s=n, although impractical at all, is still an interesting example to demonstrate this insight.
> > > > > >
> > > > > > However, as I have pointed out in the previous post, performance metrics of Byzantine-robust learning algorithms should not include “recovering the SGD rate when there are no Byzantine workers”. Emphasizing this feature would be misleading.
> > > > > >
> > > > > > In several places, the authors highlight that "existing aggregation rules fail to converge, even when no Byzantine adversaries are present” and "this is the first result establishing convergence to the optimum for heterogeneous Byzantine-robust optimization", which, in my opinion, are improper. Krum, median and geometric median are not perfect because that they are unable to handle heterogeneous data, not because they cannot recover the SGD rate.
> > > > > >
> > > > > > Hope that I have made my points clear.

---

> > > > > > > ### Author Response · Authors · 2021-12-01
> > > > > > > **A clarification request**
> > > > > > >
> > > > > > > We really appreciate the reviewer taking the time to have this deep converstation with us. This will be useful for both us and potentially the larger community who may be reading this. With that in mind, I wanted to ask for some clarification regarding this statement:
> > > > > > >
> > > > > > > > performance metrics of Byzantine-robust learning algorithms should not include “recovering the SGD rate when there are no Byzantine workers”
> > > > > > >
> > > > > > > If I understand correctly, the assertion is that the most important goal of Byzantine algorithms to be resilient to attacks (under a wide range of conditions such as heterogeneity etc.). Their convergence rate (and its optimality thereof) is secondary.
> > > > > > >
> > > > > > > We fully agree with this sentiment. If there were to be a conflict between the two goals, one should prioritize reslience over convergence.  However, for our algorithms there is no conflict between the two goals of being resilient and acheiving great rates. This, we think, is remarkable and distinguishes our work. We use the statements like "existing aggregation rules fail to converge" and "first result establishing convergence to the optimum" to draw attention to this.
> > > > > > >
> > > > > > > We will rethink how we could improve our message and would be glad to hear any suggestions on conveying our actual intentions (emphasizing that our results are great within the set of robust algorithms, not that convergence should be prioritized over robustness).

---

> > > > > > > > ### Comment · Reviewer_oF9Y · 2021-12-01
> > > > > > > > **Updates**
> > > > > > > >
> > > > > > > > I do enjoy the conversation and thank the authors for the reply again.
> > > > > > > >
> > > > > > > > The concept of Byzantine-robust optimization implies providing the worst-case guarantee. In this sense, “recovering the SGD rate when there are no Byzantine workers” is not of much relevance. As I have mentioned, for any other Byzantine-robust algorithm, if we know that there are no malicious workers, we can always switch from it to SGD to attain such a goal. But this is meaningless.
> > > > > > > >
> > > > > > > > On the other hand, emphasizing "existing aggregation rules fail to converge" and "first result establishing convergence to the optimum" could be misleading to people following this work. This is like pointing out a goal that is not worth to pursue. I would like to repeat my previous comment on this:
> > > > > > > >
> > > > > > > > In fact, I like the \delta-s tradeoff since it reveals the importance of reducing heterogeneity to Byzantine-robustness (even though the authors set s as a small value in the numerical experiments). The case of \delta=0 and s=n, although impractical at all, is still an interesting example to demonstrate this insight.

---

> > > > > > > > > ### Author Response · Authors · 2021-12-01
> > > > > > > > > **Definiton of Byzantine robustness**
> > > > > > > > >
> > > > > > > > > >The concept of Byzantine-robust optimization implies providing the worst-case guarantee.
> > > > > > > > >
> > > > > > > > > If I understood correctly the assertion is: Byzantine robustness implies being robust against large values of $\delta$ (i.e. the worst case). Stressing the performance in the setting of small $\delta$ case is bad since i) this setting is not Byzantine robustness and hence is irrelevant, and ii) redirects the research effort to an irrelevant goal.
> > > > > > > > >
> > > > > > > > > We (hopefully) understand the position of the reviewer, but must disagree. We believe there is value in providing end users with tools to set their own definition of Byzantine robustness (whether they want to apply it for the large $\delta$ case or the samall). We believe by allowing users to choose a small $\delta$ and not scarifice performance, we would encourage more wide-scale adoption. Of course, it is better to safeguard against the worst case and users should be encouraged to provide the utmost robustness. But we prefer to leave that decision to the end user. Robustness is not an absolute, but instead can be a spectrum. This is well accepted in the Differential Privacy (DP) community for e.g. where the $\epsilon$ parameter sets a privacy budget and allows meaningful tradeoffs.
> > > > > > > > >
> > > > > > > > > Having said this, we absolutely agree this is a valuable discussion to have in the paper. Comparing to DP again, companies may claim DP but set meaningless values of $\epsilon$ leading to a false sense of security. Perhaps because of this, robustness research has traditionally focused on protecting against the worst case. Nevertheless, we chose to eschew such a paradigm in our work. We believe giving more tools to the end user is a good thing, but must of course educate the readers on this.
> > > > > > > > >
> > > > > > > > > We will add this discussion to our paper - point out that traditional robustness focuses on large $\delta$, that allowing small $\delta$ could lead to a false sense of security, and elements from our current discussion. I hope this addresses the reviewer's concerns and thank them again for this discussion.

---

> > > > > > > > > > ### Comment · Reviewer_oF9Y · 2021-12-01
> > > > > > > > > > **Updates**
> > > > > > > > > >
> > > > > > > > > > The authors may misunderstand my comment. Investigating the case of small \delta is important in practice. For example, in the federated learning applications, we can roughly estimate \delta=0.05, 0.1, etc. The proposed approach is able to handle this case well through reducing the gradient heterogeneity, at the cost of tolerating a small number of Byzantine workers. This is why I voted for acceptance from the very beginning.
> > > > > > > > > >
> > > > > > > > > > But in the context of Byzantine-robust optimization, we never have the confidence that \delta=0. If it really happens and we would like to achieve exact convergence, just switch from any other Byzantine-robust algorithm to SGD. In my opinion, \delta=0 is an interesting special case to discuss, but not a valuable feature to sell.

---

> > > > > > > > > > > ### Author Response · Authors · 2021-12-02
> > > > > > > > > > > **Sorry for the misunderstanding**
> > > > > > > > > > >
> > > > > > > > > > > > the case of small $\delta$ is important in practice
> > > > > > > > > > >
> > > > > > > > > > > > we never have the confidence that \delta=0
> > > > > > > > > > >
> > > > > > > > > > > While the latter statement is true, a good way to make sure our algorithm is performing well under small $\delta$ is to looks at what happens when $\delta  \rightarrow 0$. Another way of saying this is that our algorithms *improve with decreasing $\delta$*, whereas the other algorithms do not. We will replace the claim that we recover SGD rates when $\delta =0$, with "we improve with decreasing $\delta$ unlike previous methods".
> > > > > > > > > > >
> > > > > > > > > > > Though it may be a result of misunderstanding, we believe the previous discussion on the potential pitfalls on allowing small $\delta$ is also important, and we will add that as well.

---

### Decision · Program_Chairs · 2022-01-20

**Decision:**

Accept (Spotlight)

**Comment:**

This manuscript proposes and analyses a bucketing method for Byzantine-robustness in non-iid federated learning. The manuscript shows how existing Byzantine-robust methods suffer vulnerabilities when the devices are non-iid, and describe a simple coordinated attack that defeats many existing defenses. In response, the primary algorithmic contribution is a bucketing approach that aggregates subgroups of devices before robust aggregation. This approach is also easily composed with existing Byzantine-robust methods. The manuscript includes an analysis of the performance of the proposed approach, including an information-theoretic lower bound for certain settings.

During the review, the main concerns are related to the clarity of the technical contributions, and unclear technical statements. The authors respond to these concerns and have satisfied the reviewers. After discussion, reviewers are generally strongly positive about the strength of the manuscript contributions. The authors are reminded to make the final changes agreed in the public discussion e.g., discussion of the reduction to SGD when  $\delta=0$